# TARGET-SIDE INPUT AUGMENTATION FOR SEQUENCE TO SEQUENCE GENERATION

**Shufang Xie**[1,*] **Ang Lv**[1,*] **Yingce Xia**[2]**, Lijun Wu**[2]**, Tao Qin**[2]**, Tie-Yan Liu**[2]**, Rui Yan**[1,†]
[1]Gaoling School of Artificial Intelligence, Remin University of China
[2]Microsoft Research Asia
[1]shufangxie@ruc.edu.cn, lvangupup@gmail.com, ruiyan@ruc.edu.cn
[2]{yingce.xia, lijuwu, taoqin, tyliu}@microsoft.com

## ABSTRACT

Autoregressive sequence generation, a prevalent task in machine learning and natural language processing, generates every target token conditioned on both a source input and previously generated target tokens. Previous data augmentation methods, which have been shown to be effective for the task, mainly enhance source inputs (e.g., injecting noise into the source sequence by random swapping or masking, back translation, etc.) while overlooking the target-side augmentation. In this work, we propose a target-side augmentation method for sequence generation. In training, we use the decoder output probability distributions as soft indicators, which are multiplied with target token embeddings, to build pseudo tokens. These soft pseudo tokens are then used as target tokens to enhance the training. We conduct comprehensive experiments on various sequence generation tasks, including dialog generation, machine translation, and abstractive summarization. Without using any extra labeled data or introducing additional model parameters, our method significantly outperforms strong baselines. The code is available at `https://github.com/TARGET-SIDE-DATA-AUG/TSDASG`.

## 1 INTRODUCTION

Sequence generation is an important class of machine learning tasks and has made great progress in recent years (Vaswani et al., 2017; Hassan et al., 2018; Zhou et al., 2020; Stiennon et al., 2020). In general, it is to generate sequences of interest based on a given source input. For example, in text summarization, the source input is a long document while the target sequence is a summary. In machine translation, the source input is a sentence in a language while the target output is a sentence in another language. Autoregressive (AR) methods, which generate tokens one by one autoregressively to leverage previously generated tokens, are widely used in most sequence generation problems (Gatt & Krahmer, 2018; Brown et al., 2020).

Data augmentation is a crucial technology for many machine learning tasks including sequence generation. The goal of data augmentation is to increase the number of training examples without incurring additional efforts of manually labeling, which could be costly or even impossible. For example, Wang et al. (2018) add random noise into input sequences, Gao et al. (2019) use a pretrained language model to synthesize pseudo input sequences, and Edunov et al. (2018) use back-translation to synthesize pseudo-pairs for machine translation. Furthermore, we list more works about the data augmentation method in Section 2.

While these works have demonstrated promising improvements, an inherent aspect of AR sequence generation is generally overlooked. As illustrated in Figure 1, in AR generation, each token is generated based on two kinds of input information: the original source inputs and previously generated target-side tokens (aka target-side inputs). While most existing data augmentation works focus on enhancing source inputs, the augmentation of target-side inputs is under-explored. As far as we

---

[*]Equal contribution.
[†]Corresponding author: Rui Yan (ruiyan@ruc.edu.cn).

know, only there are only a few works (Mihaylova & Martins, 2019; Zhang et al., 2019) weakly related to this problem, which are to address the exposure bias issue (Bengio et al., 2015) of the teacher forcing training. Those works adjust the sequence in token-level by randomly sampling target-side words from ground truth and model output. However, the token level adjustment may not fully leverage the potential of target input.

This work focuses on target-side data augmentation for sequence generation and proposes a sequence-level target-side augmentation method. In training, we first feed the ground truth target tokens to the decoder to generate the output probability distributions (over the target vocabulary) for each target position. Then these distributions are multiplied with the target-side token embeddings to build soft pseudo tokens as enhanced target-side inputs. Inspired by the success of mixup (Hongyi et al., 2018), the pseudo tokens are further mixed up with original ground truth target tokens to build synthetic target-side sequences. Next, our model is trained with both ground truth data and synthetic data.

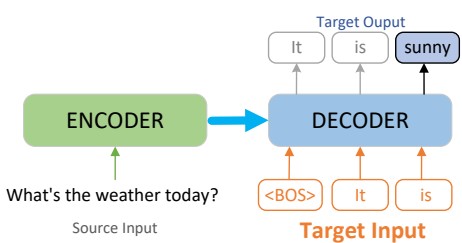

Figure 1: Illustration of autoregressive generation.

Note that we only change the target tokens for decoder input rather than target outputs or labels. Our model can benefit from the stochastic representation of target input tokens to achieve better performance. The inference process is kept the same as the conventional method. Moreover, no extra data or model parameters are required during inference. That is, we do not introduce any additional burden on model inference.

We conduct experiments on various sequence generation tasks, including dialog generation, machine translation, and abstractive summarization. On five datasets, our method outperforms standard AR training without data augmentation, as well as existing data augmentation methods.

Our main contributions are as follows:

**Problem:** We shine a light on a previously neglected problem, namely, the target-side inputs argumentation (TIA) for autoregressive sequence generation.

**Method:** We propose a new algorithm, the Soft Sequence-level Target-side Inputs Argumentation (SSTIA), which constructs synthetic target-side input tokens by generating soft pseudo-token sequences.

**Experiment:** We test our method on three sequence generation tasks with five data sets. Our method achieves good improvements over strong baselines, which reveals the importance of target-side data augmentation and the effectiveness of our algorithm. We want to call more attention from the community to design better methods for target-side data augmentation.

## 2 RELATED WORK

Data augmentation has been widely used in almost all tasks of deep learning and has been widely surveyed before (Shorten & Khoshgoftaar, 2019; Wen et al., 2021; Feng et al., 2021). In this section, we only briefly discuss works related to our method.

**Rule-based augmentation:** Some works propose to use heuristic rules to augment the source data. For example, Wang et al. (2018) randomly replace words in sentences with another word in the vocabulary, Guo et al. (2020) softly combine sequences in the training set, Shen et al. (2020) randomly erase part of a sequence to build restricted views, and Sennrich et al. (2016b) use pervasive dropout to enrich data. However, those methods target to enhance source input and do not pay attention to the target input.

**Model-based augmentation:** In this category of methods, deep learning models are used to generate the augmented data. For example, Gao et al. (2019) leverage a language model to enhance the source tokens, and Kumar et al. (2020) use BERT (Devlin et al., 2019), GPT-2 (Radford et al., 2019), and BART (Lewis et al., 2020) to generate pseudo data. For tasks with high duality, e.g., machine translation, back translation is widely used (Sennrich et al., 2016a; Edunov et al., 2018; Prabhumoye et al., 2018; Hoang et al., 2018). Usually, large-scale monolingual data is used to build pseudo training data. Similar to rule-based methods, the model-based methods do not pay much effort to the target input either.

**Knowledge distillation:** Knowledge distillation is also a widely used method for sequence generation (Hinton et al., 2015; Kim & Rush, 2016; He et al., 2019; Gou et al., 2021). In this method, teacher models are trained to generate synthetic labels for student models. However, for student models, the target input and output of the decoder are still same. On the contrary, our methods focus on changing the target input, therefore the decoder input and output are different. And we do not have any intention of model compression.

**Scheduled sampling:** Scheduled sampling methods are invented to solve the exposure-bias problem (Bengio et al., 2015; Mihaylova & Martins, 2019; Zhang et al., 2019). Although these methods also change the target input, they can only work in the token-level adaptation. In contrast, our method builds sequence-level synthetic data to improve training performance. This is an important difference between our method and scheduled sampling methods. Besides, when using sentence-level oracle, OR-NMT (Zhang et al., 2019) need full decoding and training time could be $5x$ slower.

**Reinforcement learning:** The refinement learning (RL) based sequence generation methods (Ranzato et al., 2015; Shen et al., 2016; Norouzi et al., 2016) can directly optimize the metrics that are used in test time and the target input of decoder is not always necessary. However, successfully training a RL-based sequence generation is still challenging due to its instability (Wu et al., 2018). Therefore, we focus on AR generation in this paper.

**Non-autoregressive generation:** Non-autoregressive (NAR) generation has been proposed as an alternative sequence generation paradigm, where all target tokens are generated in parallel. Although model performance has been greatly improved by many recent works (Gu et al., 2018; Qian et al., 2021; Ren et al., 2020), the AR generation is still the most common approach. Therefore, we choose to study the AR generation in this work and leave the application on NAR for future work.

**Iterative refinement:** Iterative refinement is a common way to improve performance. For example, Xia et al. (2017) proposed deliberation networks, and Lee et al. (2018) proposed to use iterative refinement in the NAR generation (Gu et al., 2018). However, these methods need multiple passes during inference time. Therefore, the iterative NAR is slower than fully NAR method (Gu & Kong, 2021). By contrast, our method only augments the data during training and has no extra inference cost. This makes our methods applicable to more scenarios where latency is critical.

## 3 OUR METHOD

In this section, we first introduce the background and notations. Next, we present the target-side data augmentation framework and our proposed method. Finally, we will discuss the relation with iterative data augmentation methods.

**Background:** Let $x$ denote the source-side input, and let $y = (y_1, y_2, \cdots, y_n)$ denote a target sequence to generate, where $n$ is the sequence length, and each $y_i$ belongs to a predefined vocabulary $\mathcal{V}$ made up of all words or sub-words. Let $D = \{(x^j, y^j)\}_{j=1}^{M}$ denote the training set with size $M$. In the AR generation, our task is to model the distribution $P(y|x)$. Specifically,

$$P(y|x) = \prod_{i=1}^{n} P(y_i|x, y_{<i}) \tag{1}$$

where $y_{<i}$ denotes the sub-sequence before $y_i$, and $y_0$ is represented by a special begin-of-sentence (BOS) token. We can see that the probability of generating $y_i$ conditions on both $x$ and $y_{<i}$.

Conventional source-side data augmentation methods can be summarized as using a function $f_{\text{aug}}$ to generate enhanced data set. Specifically, for any $(x^j, y^j) \in D$, we generate the source-side sequence by $\tilde{x}^j = f_{\text{aug}}(x^j, y^j)$, and append $(\tilde{x}^j, y^j)$ to the enhanced dataset $\tilde{D}$. The $f_{\text{aug}}$ can also solely base on $x$ or $y$. This augmentation method can be rule based (e.g., a noise function) or model based (e.g., a pre-trained language model). It may also contain some external knowledge or heuristic (Wang et al., 2018; Kobayashi, 2018; Kumar et al., 2020).

Based on the original dataset $D$ and the augmented dataset $\tilde{D}$, the training objective function is defined as follows:

$$
\begin{aligned}
\mathcal{L} &= \alpha \mathcal{L}_D + (1-\alpha)\mathcal{L}_{\tilde{D}} \\
&= -\alpha \frac{1}{M}\sum_{i=1}^{M}\log P(y^i|x^i) - (1-\alpha)\frac{1}{\tilde{M}}\sum_{i=1}^{\tilde{M}}\log P(y^i|\tilde{x}^i) \\
&= -(1-\alpha)\frac{1}{M}\sum_{i=1}^{M}\sum_{j=1}^{n_i}\log P(y_j^i|x^i,\ y_{<j}^i) - \alpha\frac{1}{\tilde{M}}\sum_{i=1}^{\tilde{M}}\sum_{j=1}^{n_i}\log P(y_j^i|\tilde{x}^i,\ y_{<j}^i),
\end{aligned}
\tag{2}
$$

The parameter $\alpha$ in Equation 2 balances the loss weight on real data $\mathcal{D}$ and synthetic data $\tilde{\mathcal{D}}$, and the $n_i$ denotes the length of $y_i$.

**Enhancing the target-side conditional input:** Different from $f_{\text{aug}}$ which focuses on enhancing the source-side input, we propose a new method that augments the previously generated target tokens, i.e., the $y_{<i}$ in equation 1. We create an augmented training data set $\breve{D} = \{(x^k, y^k, \breve{y}^k)\}_{k=1}^M$, where the $\breve{y}^k$ is the augmented sequence serving as the conditional input of the target sequence. The workflow is shown in Figure 2(a).

The loss function on $\breve{D}$ is defined as follows:

$$
\mathcal{L}_{\breve{D}} = -\frac{1}{M}\sum_{i=1}^{M}\sum_{j=1}^{n_i}\log P(y_j^i|x^i, \breve{y}_{<j}^i).
\tag{3}
$$

Some recent study (Chen et al., 2020b; Shen et al., 2020) have found that maximizing agreement between multiple augmented data that are generated from a single piece of data can improve model performance. Inspired by this, we design a consistency loss $\mathcal{L}_c$ between synthetic data and real data to stabilize the training. It is defined as

$$
\mathcal{L}_C = -\frac{1}{M}\sum_{i=1}^{M}\sum_{j=1}^{n_i}\mathcal{D}(P(\cdot|x^i,\ y_{<j}^i), P(\cdot|x^i, \breve{y}_{<j}^i)).
\tag{4}
$$

The $\mathcal{D}$ in Equation 4 can be any function to measure the divergence of two distributions, e.g., Kullback–Leibler divergence or Jensen–Shannon divergence. We found that using $\mathcal{L}_c$ can stabilize the training.

Our final loss function is defined as:

$$
\mathcal{L} = (1-\alpha)\mathcal{L}_D + \alpha\mathcal{L}_{\breve{D}} + \beta\mathcal{L}_C
\tag{5}
$$

Empirically, we found $\beta = 1$ is a good default value and used in all experiments.

**Augmented data generation:** We use model-based method to generate the augmented sequence $\breve{y}$. However, we do not need external pre-trained models like (Gao et al., 2019). Specifically,

(1) We obtain the logits $\hat{l}_j^i$ for any $j \in \{1, 2, \cdots, n_i\}$, which are the values before softmax, of all target tokens. Intuitively, $\hat{l}_j^i = \text{dec}(\text{enc}(x^i), y_{<j}^i)$, where enc and dec denote the encoder and decoder of the sequence generation model and $\hat{l}_j^i$ is a $|\mathcal{V}|$-dimensional vector. Note that we feed the groundtruth $y_{<j}$ into the decoder to obtain $\hat{l}_j^i$.

(2) We obtain the $\breve{y}_j^i$ as follows:

$$
\breve{y}_j^i = \left(\gamma\frac{\exp(\hat{l}_j^i/T)}{\sum_{v=1}^{|\mathcal{V}|}\exp(\hat{l}_j^i[v]/T)} + (1-\gamma)y_j^i\right)W_{Emb},
\tag{6}
$$

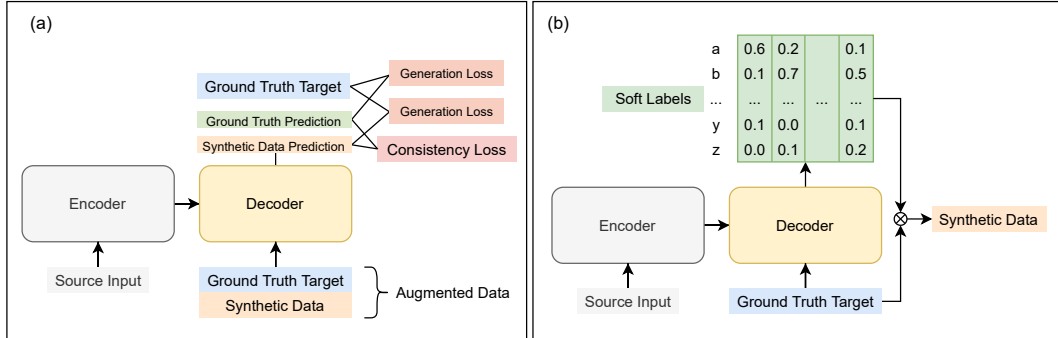

Figure 2: Illustration of our algorithm. (a) Target side data augmentation. The decoder consumes both ground truth and synthetic data as target input. (b) Augmented target data generation with soft labels. The $\otimes$ denotes mixup operation.

where (i) $W_{Emb}$ is the target word embedding with size $|\mathcal{V}| \times d$; (ii) $T$ is the softmax temperature; (iii) the square brackets $[\cdot]$ denotes the subscription of vector; (iv) $\gamma$ is the mixup ratio and we use the same value as $\alpha$ in Equation 5 for all experiments. The $\breve{y}_j^i$ is a mixture of two parts: (i) the embedding of the groundtruth $y_j^i$; (ii) a soft representation, which is the product between the target embedding and a distribution output by a sequence generation model. Empirically, we found $T$ is an important hyper-parameter and more details can be found in Section 4.4. In addition, as the modern sequence generation follows the encoder-decoder architecture , we can further reduce the augmentation cost by reusing the encoder output because it is not changed during decoding.

**Multi-round enhancement:** Similar to iterative back-translation (Hoang et al., 2018), our model can be easily extended to multiple rounds. That is, we can use the Equation 6 multiple times, where the logits $\hat{l}_j^i$ need to be replaced by the previous output. The experimental results suggest that too many iterations only give marginal extra gain. Therefore, our experiments mainly focus on single round augmentation.

## 4 EXPERIMENT

We conduct experiments on various sequence generation tasks. The experiment details and results are in the following sections. Our implementation is based on the FairSeq framework (Ott et al., 2019). We use Transformer (Vaswani et al., 2017) network architecture in all experiments with different model sizes, which are adjusted according to the data size. During training, we use Adam optimizer (Kingma & Ba, 2015) with Adam $\beta = (0.9, 0.98)$ and `invert_sqrt` learning rate scheduler. Meanwhile, we used label smoothing of value 0.1, which is the same value as Vaswani et al. (2017).

### 4.1 DIALOG GENERATION

We conduct experiments on two commonly used dialog generation data sets: the DailyDialog (Li et al., 2017) for single-turn dialog generation and Persona-Chat (Zhang et al., 2018) for multi-turn dialog generation. We follow the script of Luo et al. (2018) to pre-process the DailyDialog data, where the dialog is represented as request-response pairs. Then the first 80% pairs are used for training, the next 10% for validation, and the last 10% for test. We follow the script of Miller et al. (2017) to pre-process the Persona-Chat data, where the persona is inserted before the dialog.* We only use the *self_original* direction of Persona-Chat data in our experiments.

We use `transformer_small` configuration for DailyDialog dataset and `transformer_base` for Persona-Chat dataset, where both the encoder and the decoder consist of six layers. The $(\texttt{Embed\_Dim}, \texttt{FFN\_Embed\_Dim})$ of those configurations are $(512, 1024)$ and $(512, 2048)$, respectively. Our results are generated by beam search with beam size 5. We evaluate the BLEU (B1-B4),

---

*The data process scripts and evaluation scripts are available in Appendix.

ROUGE-L (R-L), METEOR (Met.), and CIDEr metric (Papineni et al., 2002; Lin, 2004; Banerjee & Lavie, 2005; Vedantam et al., 2015). We use `nlg-eval` (Sharma et al., 2017) to compute the metrics except for BLEU scores, which are computed by the script from Luo et al. (2018). The experimental results of DailyDialog and Persona-Chat are shown in Table 1 and Table 2, respectively.

Table 1: Experimental results on DailyDialog data set.

| Method | B1 | B2 | B3 | B4 | R-L | Met. | CIDEr |
|--------|----|----|----|----|-----|------|-------|
| Seq2Seq (Luo et al., 2018) | 12.43 | 4.57 | 2.69 | 1.84 | - | - | - |
| AEM (Luo et al., 2018) | 13.55 | 4.89 | 3.04 | 2.16 | - | - | - |
| Transformer (Our reproduce) | 14.70 | 6.13 | 4.15 | 3.13 | 17.45 | 5.36 | 18.58 |
| SeqMix (Guo et al., 2020) | 12.09 | 4.16 | 2.50 | 1.81 | 14.96 | 5.46 | 6.68 |
| SSMBA (Ng et al., 2020) | 12.96 | 4.68 | 3.03 | 2.34 | 15.69 | 4.43 | 10.23 |
| OR-NMT (Zhang et al., 2019) | 16.18 | 7.94 | 6.13 | 5.21 | 18.86 | 6.78 | 38.94 |
| **SSTIA$_1$ (Ours)** | **17.53** | **9.32** | **7.54** | **6.70** | **20.32** | **8.64** | **53.57** |

In Table 1 and Table 2, our method is first compared with conventional methods. In addition, we use recent data augmentation methods (e.g., SeqMix and SSMBA) as baselines. We also reproduce the results on Transformer and OR-NMT with sentence oracle. The subscript 1 of our method means we use single round augmentation.

From Table 1 we can see that our method is much better than existing works on all metrics, especially previous works of data augmentation. On all metrics mentioned above, we achieved propitious improvement over the Transformer baseline, especially for BLEU-4 (+3.57), R-L (+2.87), Met. (+3.28), and CIDEr (+34.99). The scores are also better than previous data augmentation methods'. We can witness the similar improvement in Table 2, especially on the R-L (+0.93) and the CIDEr (+0.73) metric. Furthermore, following Liu et al. (2020b), we conducted the human evaluation on Persona-Chat data set, and the average score is improved from 1.18 to 1.49, which verified the quality improvement of our method. More details about human evaluation is in Appendix.

Table 2: Experimental results on Persona-Chat dataset.

| Method | B1 | B2 | B3 | B4 | R-L | Met. | CIDEr |
|--------|----|----|----|----|-----|------|-------|
| Transformer (Tay et al., 2021) | - | - | - | 3.20 | 13.38 | 5.89 | 18.94 |
| Transformer (Our reproduce) | 18.33 | 8.28 | 4.99 | 3.44 | 19.27 | 8.13 | 20.42 |
| SeqMix (Guo et al., 2020) | 18.11 | 8.30 | 3.43 | 2.83 | 18.98 | 8.32 | 20.26 |
| SSMBA (Ng et al., 2020) | 18.69 | 7.07 | 3.90 | 2.72 | 19.28 | 8.30 | 11.66 |
| OR-NMT (Zhang et al., 2019) | 18.69 | 8.34 | 4.97 | 3.42 | 19.80 | 8.43 | 20.14 |
| **SSTIA$_1$ (Ours)** | **19.03** | **8.68** | **5.21** | **3.55** | **20.20** | **8.43** | **21.15** |

## 4.2 MACHINE TRANSLATION

On machine translation task, we conduct experiments on two widely used data sets: the IWSLT'14 English (EN) $\leftrightarrow$ German (DE) data set (Cettolo et al., 2014), which is a small scale data set with $140k$ sentence pairs, and the WMT'14 EN $\rightarrow$ DE dataset (Bojar et al., 2014), which is a larger one with $4.5M$ sentence pairs. We use the scripts from Ott et al. (2019) to process data. For IWSLT'14 EN↔DE, all words are lowercased and tokenized, and for WMT'14, all sentences are kept as original case then tokenized. Then the tokenized sentences are processed by BPE (Sennrich et al., 2016d) with $10k$ and $40k$ steps, respectively. We use `transformer_small` for IWSLT'14 and `transformer_base` for WMT'14, where the details are described in Section 4.1. The subscript 1 or 2 of our method means the number of rounds used in augmentation. Our results are generated by beam search using beam size 5. We compute the BLEU score by the Moses script (Koehn et al., 2007) with the same tokenizer used by previous works.

The experimental results are summarized in Table 3 and Table 4, respectively. Our baselines include standard AR training without data augmentation, other Transformer variants, and previous works about data augmentation. Compared with Transformer without data augmentation, our method achieves 1.65-point BLEU gain on IWSLT'14 EN→DE direction, 2.36-point on DE→EN direction,

and 1.8-point improvement on WMT'14 EN→DE translation (significance level 0.001). Besides, our results are better than other methods with complex architecture modification (the first group in tables) and source-side data augmentation (the second group in tables). These numbers clearly show the importance of target-side data augmentation and the potency of our algorithm.

Moreover, we added an extra round of data augmentation on the IWSLT'14 data set, and the results are in the last row of Table 3. We can see that the BLEU scores can be further boosted while the improvement is smaller than the first round's. This phenomenon is consistent with other data augmentation methods, e.g., iterative back-translation (Hoang et al., 2018).

Finally, we combine our method with previous data augmentation method, i.e., back translation. More specifically, without using extra monolingual data, we translate the target sentences in the training set using a backward translation model. When jointly using BT and our method, the BLEU score can be boosted to 37.17, which is higher than using any of the methods solely. The result exemplifies that our method is complementary to other data augmentation methods.

Table 3: Experimental results on IWSLT English ↔ German translation task.

| Method | EN→DE | DE→EN |
|---|---|---|
| Transformer (Vaswani et al., 2017) | 28.57 | 34.40 |
| DynamicConv (Wu et al., 2019) | 28.70 | 35.00 |
| MAT (Fan et al., 2020) | 29.90 | 35.59 |
| Adversarial training (Wang et al., 2019) | 28.43 | 35.18 |
| WordDrop (Sennrich et al., 2016c) | 29.20 | 35.60 |
| SCA (Gao et al., 2019) | - | 35.78 |
| SwitchOut (Wang et al., 2018) | 29.00 | 35.90 |
| SSMBA (Ng et al., 2020) | - | 36.10 |
| SeqMix (Guo et al., 2020) | 29.50 | 36.20 |
| Back Translation (Our reproduce) | - | 36.38 |
| Mixed Representations (Wu et al., 2020) | 29.93 | 36.41 |
| **SSTIA$_1$ (Ours)** | 30.14 | 36.47 |
| **SSTIA$_2$ (Ours)** | **30.22** | **36.76** |
| **SSTIA$_1$ + Back Translation** | - | **37.17** |

Table 4: Experimental results on WMT'14 English→German translation task.

| Method | BLEU |
|---|---|
| Transformer (Vaswani et al., 2017) | 27.3 |
| Admin (Liu et al., 2020a) | 27.9 |
| Evolved Transformer (So et al., 2019) | 28.4 |
| Weighted Transformer (Ahmed et al., 2017) | 28.4 |
| Adversarial Training (Wang et al., 2019) | 28.4 |
| OR-NMT (Zhang et al., 2019) | 28.5 |
| Transformer+Cutoff (Shen et al., 2020) | 29.1 |
| **SSTIA$_1$ (Ours)** | **29.1** |

## 4.3 ABSTRACTIVE SUMMARIZATION

We conduct experiments on the CNN/DM (Hermann et al., 2015) news summary dataset. To measure the effectiveness of our method with pre-trained models, We fine-tune the BART (Lewis et al., 2020) model on the CNN/DM dataset. Our experiments are based on `BART.Large` configuration. We use the scripts from Lewis et al. (2020) to process the CNN/DM data and the `files2rouge` tool to evaluate the ROUGE-1 (R-1), ROUGE-2 (R-2), and ROUGE-L (R-L) score. We report both scores and 95% confidence intervals. Our results are shown in Table 5.

In addition to BART (Chen et al., 2020a), we also compare our method with previous works on this task and the scheduled sampling method OR-NMT with sentence-level Oracle. From the numbers in Table 5, our method significantly improves the ROUGE score of the BART model with 0.6-point R-1 and 0.7-point R-L and is better than other architecture changes or fine-tuning techniques.

Table 5: Experimental results on CNN/DM dataset.

| Method | R-1 | R-2 | R-L |
|---|---|---|---|
| BART (Lewis et al., 2020) | 44.16 | 21.28 | 40.90 |
| BART + OR-NMT (Zhang et al., 2019) | 44.01 | 21.26 | 40.81 |
| PEGASUS (Zhang et al., 2020) | 44.17 | 21.47 | 41.11 |
| ERNIE-GEN (Xiao et al., 2020) | 44.02 | 21.17 | 41.26 |
| ProphetNet (Qi et al., 2020) | 44.20 | 21.17 | 41.30 |
| BART + R3F (Aghajanyan et al., 2021) | 44.38 | **21.53** | 41.17 |
| **BART + SSTIA$_1$ (Ours)** | **44.76** | 21.46 | **41.57** |
| 95% confidence interval | (44.54, 44.95) | (21.23, 21.68) | (41.36, 41.77) |

Moreover, our results are better than the scheduled sampling methods OR-NMT on sentence-oracle, demonstrating our augmentation method's effectiveness.

## 4.4 ABLATION STUDY

**Study of the parameter $\alpha$:**  $\alpha$ is a key hyper-parameter of our system, which controls the augmented data weight and mix-up ratio. To study the effect of parameter $\alpha$, we conduct experiments on the IWSLT'14 En $\rightarrow$ De data set. The Table 6 and Table 10 provide the results of this ablation study. In addition to a fixed value, we also explore two value schedule functions that change the $\alpha$ by epoch: (S1) $\alpha = \min(1, 1.5/\sqrt{\log(\text{EPOCH}) + 1})$; (S2) $\alpha = (\text{EPOCH} - 55)^2/3000 + 0.027(\text{EPOCH} < 55)$. We have more results on other datasets and explanations about the design intuition of these two formulas in the Appendix.

First, we can observe that using a moderate value gives the best results. If the value is too high, e.g., 0.9, the results are negatively affected. For the scheduled approach, different from previous works, we did not notice significant improvement. These results may suggest that a fixed value is already good enough for our algorithm.

Table 6: Ablation study of $\alpha$ on IWSLT'14 dataset.

| $\alpha$ | 0.1 | **0.3** | 0.4 | 0.5 | 0.7 | 0.9 | S1 | S2 |
|---|---|---|---|---|---|---|---|---|
| **BLEU(test)** | 36.43 | **36.47** | 36.43 | 36.40 | 36.22 | 36.19 | 36.00 | 36.38 |
| **BLEU(dev)** | 36.75 | 36.78 | 36.74 | 36.71 | 36.68 | 36.53 | 36.49 | 36.57 |

**Study of the consistency loss:**  We conduct experiments on the IWSLT'14 En $\rightarrow$ De data set to study the importance of consistency loss by changing the values of hyper-parameter $\beta$. As can be seen from Table 7, when the consistency loss is not used ($\beta = 0$), the BLEU score is improved by 0.9 point over the baseline without augmentation, which is lower than our best result. Meanwhile, the default value $\beta = 1$ gives the best result. When the value is too high or too low, the BLEU score will be negatively affected.

Table 7: Ablation study of $\beta$ in loss function.

| $\beta$ | 0 | 0.6 | 0.8 | **1** | 2 | 3 |
|---|---|---|---|---|---|---|
| **BLEU(test)** | 35.28 | 36.03 | 36.29 | **36.47** | 36.32 | 36.30 |
| **BLEU(dev)** | 35.96 | 36.39 | 36.57 | **36.78** | 36.53 | 36.46 |

**Study of the softmax temperature $T$:**  The softmax temperature $T$ is an important hyper-parameter of our algorithm. Empirically, we find a linear correlation between the best $T$ value and vocabulary size. In Figure 3, for all data sets we use, we show the vocabulary size on the x-axis and the best $T$ value on the y-axis (Detailed numbers are in Appendix). As the line chart shows, the best $T$ value linearly decreases with the increase in vocabulary size. We suspect the reason is that the high temperature will smooth the probability distribution. Therefore, we suggest adjusting this parameter according to the vocabulary size to achieve the best results.

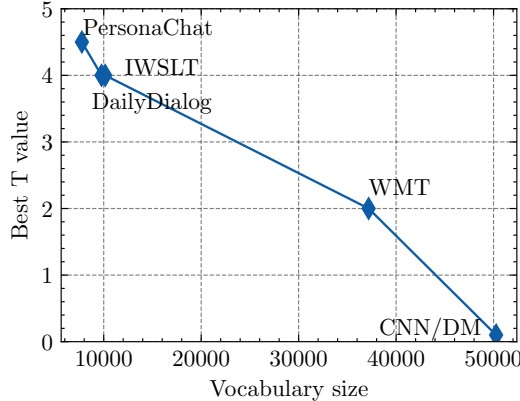

Figure 3: The best temperature on different vocabulary size.

**Study of the divergence $\mathcal{D}$:** Theoretically, we can use any function $\mathcal{D}$ in Equation 4 to measure the divergence between two distributions. We empirically choose Kullback–Leibler divergence ($\mathcal{D}_{KL}$) or Jensen–Shannon divergence ($\mathcal{D}_{JS}$) and conduct experiments on the IWSLT'14 EN$\rightarrow$ DE data set. We explore both single round augmentation and double rounds augmentation. It can be seen from the data in Table 8 that the $\mathcal{D}_{KL}$ gives better results than JS divergence. Our intuition is that the synthetic data is not symmetric to real data therefore KL is more suitable here.

Table 8: Ablation on the choice of $\mathcal{D}$.

| $\mathcal{D}$ | Single Round | Double Rounds |
|---|---|---|
| $\mathcal{D}_{KL}$(test) | 36.47 | 36.76 |
| $\mathcal{D}_{KL}$(dev) | 36.78 | 37.19 |
| $\mathcal{D}_{JS}$(test) | 35.95 | 36.50 |
| $\mathcal{D}_{JS}$(dev) | 36.34 | 36.91 |

**Iterative data augmentation:** As described in Section 3, our method can be easily extended to iterative augmentation. We conduct experiments to evaluate the improvement on the IWSLT'14 DE$\rightarrow$EN data set, and the results are in Table 9. The zero iteration means no augmentation data is used. The numbers clearly show that the first iteration gives the most improvement, and other rounds have fewer yet positive effects on performance.

Table 9: Experimental results on iterative data augmentation.

| # Iterations | 0 | 1 | 2 | 3 |
|---|---|---|---|---|
| **BLEU(test)** | 34.40 | 36.47 | 36.76 | 36.77 |
| **BLEU(dev)** | 35.49 | 36.78 | 37.19 | 37.18 |

## 5 CONCLUSION AND FUTURE WORK

This paper studies a fundamental yet widely overlooked problem: augmenting the target input information for autoregressive sequence generation tasks. While data augmentation methods have been approved to be critical for machine learning, the study of enhancing the target-side conditional input is still limited. Meanwhile, we propose a sequence-level data augmentation method by using soft labels from the decoder output. Then the mixup method is used to generate synthetic data for model training. We conducted comprehensive experiments on three sequence generation tasks of five data sets. The experimental results suggest that augmenting target input benefits the sequence generation quality. Moreover, our method can vastly boost the model performance for these tasks. We hope this can bring more attention to this type of augmentation method.

There are many potential directions for future research. First, the investigation into other methods to augment this information will be fascinating. Second, applying such an idea to large-scale pre-training could also be substantial. Last, how to combine this method with other mono data augmentation methods, e.g., a VAE model, is also a significant challenge. Therefore, we expect more research to be done in this area in the future.

ACKNOWLEDGEMENT

We would like to thank the anonymous reviewers for their insightful comments. This work was supported by National Natural Science Foundation of China (NSFC Grant No. 62122089 and No. 61876196), Beijing Outstanding Young Scientist Program NO. BJJWZYJH012019100020098, and Intelligent Social Governance Platform, Major Innovation & Planning Interdisciplinary Platform for the "Double-First Class" Initiative, Renmin University of China. We also wish to acknowledge the support provided and contribution made by Public Policy and Decision-making Research Lab of RUC. Rui Yan is supported by Beijing Academy of Artificial Intelligence (BAAI).

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

## A   DATA PROCESS AND EVALUATION SCRIPTS

The scripts for data preprocessing and evaluation can be found in the following URLs.

**DailyDialog:** `https://github.com/lancopku/AMM`

**Persona-Chat:** `https://github.com/facebookresearch/ParlAI/tree/main/projects/personachat`

**IWSLT'14:** `https://github.com/pytorch/fairseq/blob/main/examples/translation/prepare-iwslt14.sh`

**WMT'14:** `https://github.com/pytorch/fairseq/blob/main/examples/translation/prepare-wmt14en2de.sh`

**CNN/DM:** `https://github.com/pytorch/fairseq/blob/main/examples/bart/README.summarization.md`

**Moses BLEU:** `https://github.com/moses-smt/mosesdecoder/blob/master/scripts/generic/multi-bleu.perl`

**Statistical significance:** `https://github.com/moses-smt/mosesdecoder/blob/master/scripts/analysis/bootstrap-hypothesis-difference-significance.pl`

**files2rouge:** `https://github.com/pltrdy/files2rouge`

## B  ABLATION STUDY

In Section 4.4, we mentioned two $\alpha$ schedule formula. Our intuition is that the value should decrease with the training progress, while the initial model quality may not be good, which will affect the synthetic data quality. Thus, we believed it could be better to have a warm-up stage when the model only learns from real data. Empirically, we find that the performance of the model with almost all hyper-parameters do not increase any more after the 50th epoch. That's the reason that we choose 55 in S2. However, compared with a fixed value, we did not observe improvement. Therefore, we kept a fixed value in all other experiments.

Figure 4 contains the plots of these two functions.

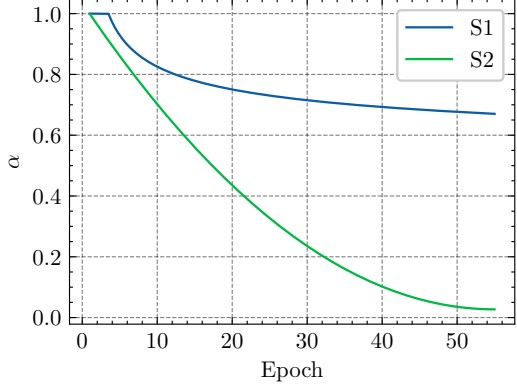

Figure 4: The $\alpha$ schedule function.

Besides, due to the limited pages, we put more albation results in Table tables 10 to 14.

## C  STUDY OF ITERATIVE AUGMENTATION

We show the training curve of IWSLT'14 dataset for different iteration round in Figure 5.

## D  TRAINING CURVE ABOUT THE CONSISTENCY LOSS

We show the training curve w/ and w/o consistency loss in Figure 6. As it shown in the figure, without consistency loss, the model will converge slower and the performance is worse.

## E  HUMAN EVALUATION DETAILS

To verify the effectiveness of out method on dialogue generation and summarization task, we recruited 5 evaluators to judge on 100 samples in blind testing. For each response, each evaluator needs to give a score from 1 (low quality) to 4 (high qualiy). The score details are shown in Table 15.

Table 10: Ablation study of $\alpha$ on Persona-Chat dataset(dev).

| $\alpha$ | BLEU-1 | BLEU-2 | BLEU-3 | BLEU-4 |
|---|---|---|---|---|
| 0.3 | 18.78 | 8.57 | 5.14 | **3.55** |
| **0.4** | **19.03** | **8.68** | **5.21** | **3.55** |
| 0.5 | 18.99 | 8.64 | **5.21** | **3.55** |
| 0.6 | 18.64 | 8.53 | 5.10 | 3.51 |

Table 11: Ablation study of softmax temperature on IWSLT'14 dataset.

| T | 0.5 | 1 | 2 | **4** | 6 | 10 |
|---|---|---|---|---|---|---|
| BLEU | 36.01 | 36.31 | 36.28 | **36.47** | 36.41 | 36.23 |

Table 12: Ablation study of softmax temperature on Persona-Chat.

| Temperature | BLEU-1 | BLEU-2 | BLEU-3 | BLEU-4 |
|---|---|---|---|---|
| 2.0 | 18.26 | 8.46 | 5.11 | 3.50 |
| 4.0 | 17.92 | 8.34 | 5.06 | 3.49 |
| **4.5** | **19.03** | **8.68** | **5.21** | **3.55** |
| 5.0 | 18.44 | 8.37 | 4.98 | 3.45 |
| 6.0 | 18.27 | 8.35 | 5.04 | 3.48 |

Table 13: Ablation on the choice of $\mathcal{D}$ on dialog(PersonaChat).

| $\mathcal{D}$ | B1 | B2 | B3 | B4 | R-L | Met. | CIDEr |
|---|---|---|---|---|---|---|---|
| $\mathcal{D}_{\text{KL}}$ | 19.03 | 8.68 | 5.21 | 3.55 | 20.20 | 8.43 | 21.15 |
| $\mathcal{D}_{\text{JS}}$ | 18.61 | 8.49 | 5.08 | 3.50 | 19.71 | 8.43 | 21.10 |

Table 14: Ablation on the choice of $\mathcal{D}$ on summarization.

| $\mathcal{D}$ | ROUGE-1 | ROUGE-2 | ROUGE-L |
|---|---|---|---|
| BART+OURS($\mathcal{D}_{\text{KL}}$) | 44.76 | 21.46 | 41.57 |
| BART+OURS($\mathcal{D}_{\text{JS}}$) | 44.59 | 21.42 | 41.43 |

Table 15: Human evaluation distribution and average score.

| Task | System | 1 | 2 | 3 | 4 | Average Score |
|---|---|---|---|---|---|---|
| Dialog | Baseline | 85.70% | 11.68% | 1.94% | 0.68% | 1.175 |
| | Ours | 62.98% | 27.90% | 6.49% | 2.63% | 1.487 |
| Summarization | Baseline | 2.27% | 14.77% | 34.09% | 48.86% | 3.300 |
| | Ours | 0.00% | 9.09% | 32.95% | 57.95% | 3.490 |

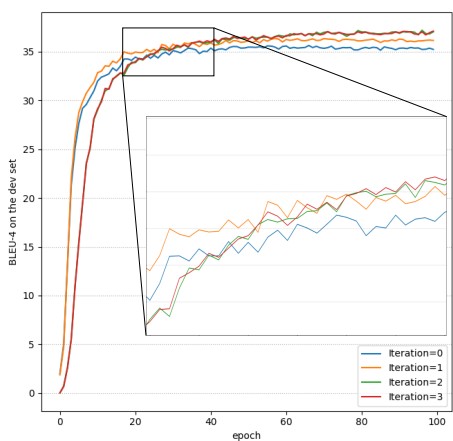

Figure 5: Study of the iterative augmentation on IWSLT.

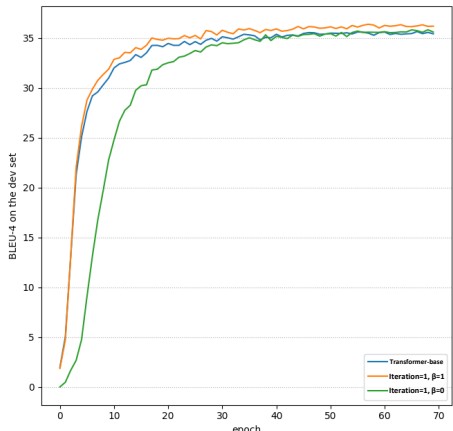

Figure 6: BLEU-4 on the validation set during training.

