# OpenReview forum: "Target-Side Input Augmentation for Sequence to Sequence Generation"
_ICLR.cc/2022/Conference — ICLR 2022 Poster_

### Official Review · Reviewer_k7SM · 2021-10-31

**Correctness:** 2
**Technical Novelty And Significance:** 2
**Empirical Novelty And Significance:** 3
**Recommendation:** 5
**Confidence:** 4

**Main Review:**

Strengths:
1. The paper performs comprehensive experiments on three different generation tasks (i.e. dialog generation, machine translation, and abstractive summarization) and demonstrates improvements over baselines.
2. Their idea of constructing the augmented data can be viewed as a fast approximation of scheduled sampling, which is technically sound.
3. The paper is easy to follow.

Weaknesses:
1. They seem to perform ablation studies and choose hyper-parameters on the test sets.
2. The paper keeps emphasizing that target-side data augmentation has been "overlooked" or "neglected" in sequence generation throughout the paper, while a lot of papers in this direction have been proposed (e.g. scheduled sampling [1], many RL-based methods [2,3,4], some heuristics [5,6], self-training [7], ...).
3. The consistency loss is unintuitive and more analyses are required. If the model is trained to output the same targets on the two views, we are already encouraging the consistency between them, so why would an extra consistency loss be helpful here?
4. The comparisons with baselines are not fair in some cases. The author(s) state that "for a fair comparison with previous works, we compute the BLEU score by the Moses script ", but the scores obtained by this script depend on the tokenizer, which can be different in different papers and SacreBLEU (https://github.com/mjpost/sacrebleu) is recommended for a truly fair comparison.



[1] Bengio et al., Scheduled sampling for sequence prediction with recurrent neural networks, NeurIPS 2015.

[2] Norouzi et al., Reward augmented maximum likelihood for neural structured prediction, NeurIPS 2016.

[3] Ranzato et al., Sequence level training with recurrent neural networks, ICLR 2016.

[4] Shen et al., Minimum risk training for neural machine translation, ACL 2016.

[5] Sennrich et al., Edinburgh neural machine translation systems for WMT 16, WMT 2016.

[6] Wang et al., SwitchOut: an Efficient Data Augmentation Algorithm for Neural Machine Translation, EMNLP 2018.

[7] He et al., Revisiting Self-Training for Neural Sequence Generation, ICLR 2020.



**Summary Of The Paper:**

This paper focuses on sequence generation tasks and proposes to perform data augmentation on the target side. During training, given an input-output pair ($x$, $y$), their model first uses teacher-forcing to get the output probability $p_y$ and then gets an augmented pair ($x$, $\tilde{y}$) by using "mixup" to mix the embeddings of $y$ and $p_y$. They also propose to encourage the consistency of predictions between ($x$, $y$) and ($x$, $\tilde{y}$) by minimizing the KL divergence of their output probabilities.

They experiment on three sequence generation tasks and demonstrate improvements over baselines.

**Summary Of The Review:**

While I think their idea of augmenting the dataset is interesting and it does not introduce any inference burden, based on the weaknesses I mentioned above, I'm leaning towards a rejection of the paper.

---

> ### Author Response · Authors · 2021-11-21
> **Response to Reviewer k7SM**
>
> Thanks for your valuable feedback on our paper! Here are the answers to your questions.
>
> > 1. Question about hyperparameter selection and ablation study
>
> Although we only report the results on the test sets, all our checkpoint selection and hyperparameter selection are based on the performance of the validation set. In addition, we add the training curve in the Appendix of the revised version to better demonstrate the effectiveness of our method.
>
>
> > 2. The claims about target-side data augmentation
>
> We feel sorry for the confusion about this statement and feel grateful for providing these related works. We agree that these papers discuss the modification of target sequence, however, there is still a key difference between our method and theirs. Our point is that we can enhance the target sequence in the decoder input while keeping the output. This means the input and output of the decoder are not the same and the model can benefit from the enhanced representation. However, in most previous works (except scheduled sampling), the decoder input and the output are the same regardless of shift operation due to AR generation. Compared with scheduled sampling, our sequence level soft token could be more generated faster and achieve better results. Meanwhile, in the experiments, we have already compared with other data augmentation methods. For the paper we missed, we have added the discussion with them in the revised version.
>
> > 3. Question about the motivation for consistency loss
>
> The consistency loss is motivated by recent advances in utilizing augmented data. For example, in SimCLR[1] and Cut-Off[2], the consistency loss is computed from two different augmented data to maximize the agreement. Instead of using two augmented data, we directly compare the output of synthetic data and real data to improve the model performance. Empirically, we find this will make the model converge faster and achieve better results. We will revise the paper to better explain this motivation.
>
> [1] Chen, Ting, et al. "A simple framework for contrastive learning of visual representations." International conference on machine learning. PMLR, 2020.
>
> [2] Dinghan Shen, et al.  "A simple but tough-to-beat data augmentation approach for natural language understanding and generation", 2020
>
> > 4. Question about the BLEU score computation
>
> We agree that the tokenizer is critical for BLEU score. For the multi-bleu, we follow the convention that the sentences are tokenized by the script `mosesdecoder/scripts/tokenizer/tokenizer.perl`, which is widely used in previous studies including the baselines we mentioned. Therefore, we believe it is a fair comparison because they are all using the same tokenizer and the same BLEU evaluation script. We use multiBLEU because we follow the previous baseline methods. In terms of sacreBLEU, our result is 27.45, which is one point higher than the standard Transformer base model.
>
> -----
> We hope that these answers can address your questions. If you have future suggestions or questions about our paper, we will feel very happy to share more responses.

---

> > ### Comment · Reviewer_k7SM · 2021-11-22
> > **The response does not address my concerns**
> >
> > Thanks to the author(s) for the response! However, the response still does not address my concerns:
> >
> > 1. Please include the model performance on the dev sets to support this claim.
> >
> > 2. I don't think "enhancing the target sequence" is well-defined and I'm not convinced that target-side data augmentation can be narrowed down to methods whose "input and output of the decoder are different." All the methods that I mentioned can perform augmentation on the target side and the paper needs a better way of discussing the differences between their proposed method and previous methods.
> >
> > 3. The consistency loss makes sense for unlabeled data. However, for labeled data, the model is already trained to maximize the probability of the same ground-truth targets on both views, thus the consistency has already been optimized. Why would an additional consistency loss be helpful?
> >
> > 4. Using multi-bleu.perl to compute the BLEU scores makes the comparison inherently unfair, which has been explicitly stated in the multi-bleu.perl script. Even if you are using the same tokenizing script in moses with previous work, different methods can use different tokenization arguments. I'm not saying that you cannot list all the BLEU scores computed by multi-bleu.perl in one table, but that you shouldn't say that the comparison is fair if you are using multi-bleu.perl.

---

> > > ### Author Response · Authors · 2021-11-23
> > > **Response to Reviewer k7SM**
> > >
> > > Thank you for the further comments!
> > >
> > > > Q1: Results on the validation set
> > >
> > > Here are the results of IWSLT for different $\alpha$.
> > >
> > > |$\alpha$| 0.1 | 0.3|0.4|0.5|0.7|0.9|s1|s2|
> > > |-----------|------|-----|----|----|----|----|----|---|
> > > |dev| 36.75|36.78|36.74|36.71|36.68|36.53|36.49|36.57|
> > > |test| 36.43|36.47|36.43|36.40|36.22|36.19|36.00|36.38|
> > >
> > >
> > > From the table, we can see that (1) the $\alpha$ arounds 0.3 have the best result. (2) scheduled $\alpha$ do not have improvement over the cost values.
> > >
> > > > Q2: Better define the target-input augmentation
> > >
> > > The formal definition is in the Section 3 of our paper. Briefly, an standard AR generation model approximate the distribution $P(y|x) = \Pi_{i=1}^n P(y_i|x, y_{<i})$, where the $y_{<i}$ is the target input to the decoder. However, in our setting, we model the distribution $P(y|x) = \Pi_{i=1}^n P(y_i|x, \breve{y}_{<i})$, where the $\breve{y}$ is an augmented representation of $y$. Note that the target output is still $y$ not $\breve{y}$. Conventional methods, e.g., knowledge distillation, self-training, back-translation, etc., will create new training data that include new (x, y) pairs. During training, we can model the new training data using either AR training target or ours.  For Scheduled sampling, it can also change target input. But these methods usually make the training much slower while our sequence level generation is faster. In addition, we achieved better results compare with these methods.
> > >
> > > Meanwhile, our method is stackable with these augmented methods. For example, we have a new experiment that combine BT with our method. We do not use extra mono data but simply back translate the training data. The results are:
> > >
> > > | Method | BLEU|
> > > |:----------|:-------:|
> > > | Transformer| 34.40|
> > > |+Back-Translation (BT) | 36.38	|
> > > |+Ours  | 36.47 |
> > > |+BT, +Ours | 37.17|
> > >
> > > The results indicate that using our method on the BT augmented data can further improve the performance.
> > >
> > > > Q3: Motivation about consistency loss
> > >
> > > The consistency loss is inspired by the one widely used in multi-view learning. We agree that most of them are used for unlabeled data, empirically we find it is also useful in our framework.
> > > Our intuition is that the synthetic data could be more complex and more diverse compared with real data. Therefore, it could be helpful to let the output of synthetic data get close to the output of real data. From this perspective, It is also similar to the idea used in knowledge distillation to create easy-to-learn labels. Future more, we believe that exploring consistency of real data and augmented data is a general problem of machine learning and could be an important study. However, that may much beyond the scope of this paper.
> > >
> > > > Q4: Question about fair use of multi-bleu
> > >
> > > We agree that tokenization parameters will affect the results. We use the scripts provided by fairseq (https://github.com/pytorch/fairseq/tree/main/examples/translation) without changing any tokenization parameters. Therefore, if other work also uses the same scripts or same tokenizer, the comparison will be fair. We will revise the paper to make it clearer.

---

> > > > ### Comment · Reviewer_k7SM · 2021-11-24
> > > > **Response**
> > > >
> > > > 1. Please include the dev set results for *all of your ablation studies (i.e. Table 6, 7, 8, 9) in your paper*. I don't know why the author(s) only post the results of one table after I explicitly asked twice.
> > > >
> > > > 2. Rephrasing "target-side data augmentation" to "target-side input augmentation" addresses my concern to some extent, but please make the changes throughout the paper accordingly. Claims such as "previous methods overlook the target-side augmentation" are inaccurate.
> > > >
> > > > 3. I've asked the authors twice about this motivation but I still haven't got a direct answer, thus I'll rephrase the question: if our target is $t$ and we are minimizing both the distance between $a$ and $t$ and the distance between $b$ and $t$, then $a$ and $b$ are already being pushed towards the same target $t$, why would we need another loss to minimize the distance between $a$ and $b$?
> > > >
> > > > 4. You still cannot make sure the tokenizers are consistent across different research groups. As I stated in my previous comment, I don't have objections towards listing all the BLEU scores computed by multi-bleu.perl in one table, but you should remove the "fair comparison" claim.

---

> > > > > ### Author Response · Authors · 2021-11-25
> > > > > **Response to Reviewer k7SM**
> > > > >
> > > > > Thank you for the further comments!
> > > > >
> > > > > > Q1: More results about the dev results
> > > > >
> > > > > We feel sorry that we didn’t understand that all dev results are required. For table 7, the scores are already on dev set because we follow the convention of [1].
> > > > >  In addition to Table 6, here are the dev results for Table 8, 9,10.
> > > > >
> > > > > | beta      | 0     | 0.6   | 0.8   | 1     | 2     | 3      |
> > > > > |-----------|-------|-------|-------|-------|-------|--------|
> > > > > | BLEU(dev) | 35.96 | 36.39 | 36.57 | 36.78 | 36.53 | 36.46  |
> > > > > | BLEU(test) | 35.28 | 36.03 | 36.29 | 36.47 | 36.32 | 36.30  |
> > > > >
> > > > > |          | single round | double round  |
> > > > > |----------|--------------|---------------|
> > > > > | KL(dev)  | 36.78        | 37.19         |
> > > > > | KL(test) | 36.47        | 36.76         |
> > > > > | JS(dev)  | 36.34        | 36.91         |
> > > > > | JS(test) | 35.95        | 36.50         |
> > > > >
> > > > >
> > > > > | Iteration | 0     | 1     | 2     | 3      |
> > > > > |-----------|-------|-------|-------|--------|
> > > > > | BLEU(dev) | 35.49 | 36.78 | 37.19 | 37.18  |
> > > > > | BLEU(test) | 34.40 | 36.47 | 36.76 | 36.77  |
> > > > >
> > > > >
> > > > >  [1] Liu, Qian, et al. "You impress me: Dialogue generation via mutual persona perception." In Proceedings of ACL. 2020.
> > > > >
> > > > >
> > > > > > Q2: More revision to emphasize the target *input* augmentation.
> > > > >
> > > > > We have already changed the title of our paper to highlight this point, as well as refined the method name to emphasize the difference. For other places where it may still not clear, we will change them all in the paper. Because we cannot update the pdf in this phrase, we will upload the next revised version to make it more accurate later.
> > > > >
> > > > >
> > > > > > Q3: More explanations about consistency loss
> > > > >
> > > > > Sorry for the confusion.
> > > > >
> > > > > 1.       Following your notations, if the training loss of minimizing ${\rm distance}(a,t)+ {\rm distance} (b,t)$ can reach zero, using the additional term ${\rm distance}(a,b)$ is not helpful. However, due to the high non-convexity of neural networks especially for sequence generation tasks, it is almost impossible to make the training loss zero. Therefore, we believe that the additional training signal ${\rm distance}(a,b)$ is helpful too.
> > > > >
> > > > > 2.       From another perspective, knowledge distillation is often helpful to improve the performance of sequence generation tasks. Minimizing ${\rm distance}(a,b)$  shares similar inspiration to knowledge distillation, where the two modules can learn from each other so that the eventual results are expected to be improved.
> > > > >
> > > > >
> > > > > > Q4: The use of multi-bleu
> > > > >
> > > > > Thank you for the comments. We will make the revision as you suggested and remove the claim in the next version.

---

> > > > > > ### Comment · Reviewer_k7SM · 2021-11-26
> > > > > > **Thank you for the response**
> > > > > >
> > > > > > Thank the author(s) for the response. The motivation of the consistency loss still puzzles me and only showing empirical improvements may not be enough. Also, the definition of "target-side input augmentation" is still a bit strange, though it can be more accurate than the previous one, so I'd encourage the author(s) to find a better way of framing their methods. But many of my other concerns are addressed so I've increased my score.

---

### Official Review · Reviewer_ojZ7 · 2021-11-02

**Correctness:** 3
**Technical Novelty And Significance:** 3
**Empirical Novelty And Significance:** 2
**Recommendation:** 6
**Confidence:** 4

**Main Review:**

Strengths:
1. Comprehensive Experiments on three different domains - Dialog, Summarization, and Machine Translation.
2. Data Augmentation doesn't require access to other resources than the primary model

Concerns/Questions:
- Renaming Sequence Generation to Sequence-to-Sequence Model (Or Encoder-Decoder) Generation since the domain is restricted to that in this work. Although sequence generation through Language Models might use a similar scheme, this work hasn't been experimented with.
- Section 2: Comparison with KD: Our method focus on target input, not target output. This part is not very clear. If target output changes, doesn't it also affect target side inputs?
- Why is training not stabilized with just equation (3). How do you define stabilization of training? Why does consistency loss make the training stable?
- Number of turns in Tables 1, 2?
- Is the number of training passes equal for all the methods in Table 3. If not, can it be standardized? It seems like the "Ours" model is trained through more passes of the dataset (+ augmentation)
- Table 4 and 5: Are the results statistically significant? Would you please report those values?
- Section 4.3: we select the BART to initialize our model, and fine-tune on the CNN/DM -> We fine-tune the BART model on the CNN/DM dataset
- Section 4.4 [And consequently Table 6-9]: Ablation Study section is not an ablation study, which ideally involves removal of proposed changes [in terms of modeling changes or objective functions]. It seems more like hyperparameter tuning.
- Section 4.4: How is the value of 55 chosen in $\alpha$ tuning?
- Table 10: What happens when \beta = 0 ? Does the model empirically display stability issues (as mentioned earlier in the motivation for consistency loss)?
- Study of divergence -> What is the intuition of doing this? Why does KL work better than JS? JS is symmetric, so shouldn't it help?
- Improper use of "significantly boost the model" performance in conclusion when no statistically significant studies have been conducted.
- Generative tasks require human judgments for proper analysis. Kindly evaluate that.
- Future Work - "how to combine this method with unlabeled data is also a significant challenge" - Why is that so? I believe that the current model also does not utilize any labeled data. Would you please correct me if I am mistaken?


Typo/Correction needed:
- Section 2: Comparison with Iterative Refinement or NAR strategies: Claim about inference cost is incorrect since autoregressive methods are known to be slower than NAR.
- Equation (3) -> There should be a negative sign in the loss function
- Capitalize: Rouge -> ROUGE at all places

**Summary Of The Paper:**

This paper experiments with a target-side sequence-level data-augmentation scheme for sequence-to-sequence generation tasks. The primary contribution of this work is an algorithm that leverages model-outputs to construct pseudo-target-side tokens (and consequently pseudo-sequences) for augmentation. It is done by incorporating soft-embeddings in the encoder-decoder model, a standard cross-entropy loss function supercharged with a consistency loss objective. One of the critical highlights is that this work doesn't require any external model for performing data augmentation. A comprehensive set of experiments on three sequence generation tasks are conducted to highlight the effectiveness of using such an approach for data augmentation.

**Summary Of The Review:**

The work proposes a data-augmentation strategy using soft labels during training instead of hard labels as in scheduled sampling.  One of the key highlights is that this work doesn't require any external model for performing data augmentation. However, the need for consistency loss is unclear, and using this model is not observable. While the paper reports incremental results, the lack of significance testing makes it difficult to comprehend if the results are not by-chance. The ablation study section is not an ablation study but hyper-parameter tuning. Additionally, the lack of experimental details regarding the number of examples seen by each model during training (number of epochs * samples) makes it unclear if the incremental results are due to more extended training or the overall augmentation strategy.

In its current form, and based on the reasons stated above, I would recommend a weak reject for this paper.

---

> ### Author Response · Authors · 2021-11-21
> **Response to Reviewer ojZ7**
>
> Thanks for your valuable feedback on our paper! Here are the answers to your questions.
>
> > 1. Question about the paper naming.
>
> We agree that using sequence-to-sequence generation could be more accurate because we don’t have experiments with language models. We have updated it in the revised version. Meanwhile, our model can be easily extended to LM tasks.
>
> > 2. Question about the difference with KD
>
> We feel sorry for the confusion here. What we want to emphasize is that we enhance the target input of the decoder. Although KD will generate a new synthetic target sequence, the input and output of the decoder are still the same (except for the shift due to AR decoding). However, our decoder input is the enhanced sequence while the decoder output is the ground truth sequence.
>
> > 3. Question about the motivation of the consistency loss
>
> We feel sorry about the inaccurate explanation about the consistency loss. The consistency loss is motivated by the recent advances in utilizing augmented data. For example, in SimCLR[1] and CutOff[2], the consistency loss is computed from two different augmented data to maximize the agreement. Instead of using two augmented data, we directly compare the output of synthetic data and real data to improve the model performance.  Empirically, we find this makes the model converge faster and achieve better results. We will revise the paper to better explain this motivation.
>
> [1] Chen, Ting, et al. "A simple framework for contrastive learning of visual representations." International conference on machine learning. PMLR, 2020.
>
> [2] Dinghan Shen, et al.  "A simple but tough-to-beat data augmentation approach for natural language understanding and generation", 2020
>
> > 4. Question about the number of turns in Table 1, 2
>
> We use single-round augmentation in most of the experiments unless it is explicitly described that multi-round augmentation is used. To be more explicit, we add the subscript in the revised version to reflect the number of augmentation turns.
>
> > 5. Question about the training passes
>
> We control the training passes by setting the maximum number of updates. In addition, we add the training curve to further demonstrate the effectiveness of our method. These plots can be found in the Appendix of the revised version.
>
> > 6. Question about the description about using BART
>
> We feel grateful for your suggestion and have made the change accordingly in the revised paper
>
> > 7. Question about the ablation study and what happens if beta=0
>
> The first thing we study is augmented data. That is the reason we show results on different $\alpha$. Besides, we also study the effect of consistency loss.
> We appreciate your suggestions about figuring out what happens when beta=0 (i.e., remove the consistency loss) and we have added the new results in the revised version. We find that when beta=0, the model converges slower and the result is 35.28, which is better than baseline but worse than our best result.  This indicates that consistency loss is also important in our model.
> Furthermore, we analyze the effect of the selection of divergence function and multi-round augmentation.
>
> > 8.	Question About the choice of epoch 55
>
> We want to construct a function to keep decreasing alpha during training. Empirically, we find that the performance of the model does not increase any more after the 50th epoch. That's the reason we choose the number 55 as the max number of epochs to use.
>
> > 9.	Question about the study of divergence
>
> Although by generating augmented "pseudo tokens" based on ground truth, we adapt the model to the situation when the decoder's next input token is generated all by itself, narrowing the gap between training and inference, we still hope the output based on augmented data can resemble the output based on the ground truth. Hence, the first training round with teach forcing and the other augmented training rounds are asymmetric. Therefore, we choose KL divergence. The results also prove that KL does work better than JS.
>
> > 10. Question about the significant level
>
> For machine translation, we use the Moses script ( https://github.com/moses-smt/mosesdecoder/blob/master/scripts/analysis/bootstrap-hypothesis-difference-significance.pl ) to compute the significant level of our system compared with the AR training baseline. The P-value is 0.001. For the summarization task, we take the default output of files2rouge. The 95% confidence interval for R-1, R-2, and R-L are (44.54, 44.95), (21.23, 21.68), and (41.36, 41.77), respectively. Compared with BART, our improvement is significant.

---

> > ### Author Response · Authors · 2021-11-21
> > **Response to Reviewer ojZ7 (continued)**
> >
> > >11. Question about human evaluation results
> >
> > We conducted a human evaluation on Persona-Chat. We recruited 5 evaluators to judge on 100 samples in blind testing. For each response, each evaluator needs to give a score from 1 (low quality) to 4 (high quality). The detail scores are
> >
> > | System | 1       | 2       | 3      | 4      | Average Score |
> > |---------------|---------|---------|--------|--------|---------------|
> > | Baseline      | 85.70% | 11.68% | 1.94% | 0.68% | 1.175         |
> > | Ours          | 62.98% | 27.90% | 6.49% | 2.63% | 1.487         |
> >
> > From the results, the average score is improved from 1.175 to1.487, which verifies the quality improvement of our method.
> >
> > > 13. Question about future work on mono data
> >
> > We feel sorry for the confusion. The meaning was to investigate how to combine this method with other mono data augmentation methods, e.g., a VAE model. We have revised the sentence to make it clearer.
> >
> > > 14.Comparison with NAR method
> >
> > We feel sorry for the confusion. What we meant is that the iterative refinement is widely used in NAR, and the iterative NAR system is slower than the fully NAR system. We agree that the AR is slower than NAR. Nevertheless, the AR generation is still used more widely compared with AR. We have refined this section to make it clearer.
> >
> >  >15. Typos about loss function and ROUGE
> >
> > We feel grateful for your suggestions and have fixed them all in the revised version.
> >
> > ---
> > We hope that these answers can address your questions. If you have future suggestions or questions about our paper, we will feel very happy to share more responses.

---

### Official Review · Reviewer_vrEs · 2021-11-02

**Correctness:** 3
**Technical Novelty And Significance:** 2
**Empirical Novelty And Significance:** 2
**Recommendation:** 6
**Confidence:** 4

**Main Review:**

### Strengths:
This paper presents an approach to tackle an important problem in the field of natural language generation.
Its method is straightforward and well illustrated.
Furthermore, the paper adheres to scientific standards as the authors release their code.
It is also very much appreciated that they illuminate and evaluate their model in different text generation tasks.
Furthermore, the experimental results demonstrate certain improvements in these tasks.

### Weaknesses:
My main concern with this paper is that since the augmented target-side input is produced by the decoder conditioning on the ground truth, this method seems helpful but limited.
Generally, given the ground truth, the model produces the target text that is quite similar to its corresponding ground truth, and the discrepancy between them is subtle.
Thus it still cannot narrow the gap of decoding between training time and inference time.
The modest improvements in the NMT task and abstract summarization task also indicate the existence of this problem, and cast doubt on its effectiveness.

### Questions for the author(s):
Q1. As far as I know, in the NMT task, en-->de newstest2014, the best BLEU result for "OR-NMT", "Transformer+Cutoff", "Adversarial Training" are 28.65, 29.1, and 29.52 in their corresponding papers respectively.
Are there any mistakes?

Q2. The usage of \alpha on Eq(2) contradicts that on Eq(4)?



**Summary Of The Paper:**

Traditionally, the decoder of Seq2Seq model takes ground truth words of previous steps as input during training, while at inference, its input are those generated tokens starting from scratch.
Thus there exists discrepancy between training and inference.

This paper presents an approach of data augmentation for the input of the decoder during training.
Specifically, the authors still feed the ground truth to the decoder, and then obtain its output vocabulary distributions that are multiplied with word embedding weights to get "pseudo tokens".

The contribution of this work comes from narrowing the gap of decoding procedure between training and testing.

**Summary Of The Review:**

Overall, this work tries to tackle the important problem of language generation and proposes a data augmentation method to give soft pseudo decoder input.
As what has been mentioned in the weaknesses part of Main Review, there still exist problems to solve and need to prove the effectiveness of such method. Therefore, I'm leaning towards a weak rejection of this paper.

---

> ### Author Response · Authors · 2021-11-21
> **Response to Reviewer vrEs**
>
> Thanks for your valuable feedback on our paper! Here are the answers to your questions.
>
> >1.  The main concern about effectiveness and performance
>
> We agree that the target-side text produced by the decoder does resemble its corresponding ground truth, as you mentioned. However,  we use soft tokens as augmented data. We believe there's an important difference between our method and others.
> We understand your concern that the decoder-generated sequence may be similar to the ground truth sequence. However, the diversity of the natural language makes our method useful in the generation tasks. Even a single German word "Glücklich"  can be translated as both "lucky" or "fortunate" in English. Without doubts, the pattern could be more complicated for more complex sentences. In our method, the distribution over the vocabulary is used. Therefore, instead of a single target token, the model has a chance to learn the distribution and generate better results.
>
> When it comes to the effectiveness, compared with Transformer without augmentation, we achieved 2.36 BLEU gain on IWSLT De->En task. Meanwhile, on the summarization task, our method has a 0.6-point improvement on R-1 and R-L. The gain is much larger than the baseline systems. Furthermore, the 44.76 points R-1 score achieved the SOTA result on this metric.
>
> Furthermore, our method can be combined with other methods to achieve better performance. For example, we have a new experiment that combines our method with back translation (BT) on IWSLT'14 DE->EN.
>
> | Method | BLEU|
> |:----------|:-------:|
> | Transformer| 34.40|
> |+Back-Translation (BT) | 36.38	|
> |+Ours  | 36.47 |
> |+BT, +Ours | 37.17|
>
>
> > 2.  Question about the best BLEU results of baselines
>
> We are sorry that we used the result of "Transformer + Cutoff without JS loss", which is a little lower than their best result "29.1". We have fixed it in the revised version. Thank you for pointing out this.
> For others, because we use the "Transformer-base" setting, we report the BLEU score of "Transformer-base" from the other two baselines. For “OR-NMT” and “Adversarial Training”, as we reported in the previous version, the comparable results on WMT14 are "28.5" and  "28.4" . "28.65" and "29.52" are their results using "Transformer-big".
>
> > 3. The typos in Eq(2) and Eq(4)
>
> Thank you for finding this. We have fixed it in the revised version.
>
> ---
> We hope that these answers can address your questions. If you have future suggestions or questions about our paper, we will feel very happy to share more responses.

---

> > ### Author Response · Authors · 2021-11-28
> > **Response to reviwer vrEs**
> >
> > Dear Reviewer,
> >
> > We appreciate a lot for your insightful review comments. Do you have any further comments to our paper?
> >
> > Regards.

---

> > > ### Comment · Reviewer_vrEs · 2021-11-29
> > > **Response to the author(s)**
> > >
> > > Thank the author(s) for the response.
> > >
> > > Empirically, feeding with the "soft tokens" remains much more information than discrete tokens produced by maximizing probabilities. And it also has a better tolerance in terms of generating diverse texts.
> > >
> > > However, it still does not address my concerns. I suggest that feeding with dynamically generated "soft tokens" seems better and may achieve further improvement. In this way, the model is exposed to the setting of inference time, and it will narrow the gap between training and testing.
> > > Overall, I appreciate that this work focuses on the target side of the seq2seq model and achieves positive results. Therefore, I have increased my score.

---

### Official Review · Reviewer_iLjj · 2021-11-02

**Correctness:** 3
**Technical Novelty And Significance:** 3
**Empirical Novelty And Significance:** 3
**Recommendation:** 8
**Confidence:** 4

**Main Review:**

Strengths:
+ The proposed technique is generally applicable to various NLG tasks.
+ The paper is nicely-structured and very well-written. I like the presentation style of the paper -- a good mixture of formulas for mathematical rigor and textual/graphical explanations of the intuitions behind them.
+ Experiments span three different generation tasks, showing the general benefit of the proposed augmentation technique, with a detailed ablation study on the sensitivity of some hyper-parameters.

Weaknesses:
+ The improvements in machine translation and abstractive summarization are marginal compared to other augmentation techniques, but this is not a non-fixable issue, see the first point in my detailed feedback.
+ I imagine this augmentation method will be computationally expensive, since at every time step $O(|V|d)$ extra additions need to be done ($d$ being the number of word embedding dimensions). It will be useful to have some discussion on that. I also think the number of extra operations can be easily reduced by sparsifying the softmax distribution before doing the embedding interpolation.
+ I have some minor doubts about the real-world usefulness of the method. For example, for machine translation, isn't this doing a similar job as back-translation?

More detailed feedback/suggestions:
+ In the machine translation experiment, the authors compared their method with a series of other data augmentation methods such as WordDrop and SwitchOut, but I think they don't necessarily need to compete with each other -- how about, say, we use SwitchOut on the source side and the proposed augmentation method on the target side? Showing that the proposed method is stackable with other augmentation methods could make this work more impactful.
+ Continuing on the previous point, it will also be interesting to see if the proposed method is also stackable with back-translation when extra monolingual data is provided, especially since the authors have already considered multi-round enhancement.
+ Page 3: "It may also contains" -> "It may also contain"
+ Page 4: Equation (4) -- should there be an extra summation over $j$?
+ Page 8: I'm actually not sure why would higher temperature $T$ be helpful under smaller vocabularies. It would be nice to elaborate.

**Summary Of The Paper:**

This paper introduces a simple technique for target-side data augmentation. The high-level idea is to generate a "soft token" embedding by interpolating the target side word embeddings, with the output distribution generated by an initial model being the interpolation weight (after being adjusted by a temperature parameter $T$). The resulting training loss is in three terms: (1) original cross-entropy loss; (2) cross-entropy loss with the soft token embedding as the input; (3) a consistency loss controlling the distribution divergence of (1) and (2).

Results show that the proposed data augmentation method is very helpful for the dialog generation task, while also being somewhat helpful with neural machine translation for abstractive summarization. Ablation studies show that the weight of (1) cannot be too low, and that the optimal temperature parameter has some negative correlation with the vocabulary size.

**Summary Of The Review:**

This paper proposed a simple and novel target-side data augmentation technique that is at least helpful for some natural language generation tasks. I'm going with a weak-accept for now citing some doubts mentioned in the main review, but am happy to update with clarifications from the authors.

---

> ### Author Response · Authors · 2021-11-21
> **Response to Reviewer iLjj**
>
> Thanks for your valuable feedback on our paper! Here are the answers to your questions.
>
> > 1. Question about model performance when stacked with other methods.
>
> We appreciate the idea of combining our method with other data augmentation methods. Currently, we have finished the experiment that combines our method with back-translation (BT) on IWSLT'14 DE->EN. More specifically, without using extra monolingual data, we use a reverse model to back translate target sentences in the training set. The results are:
>
> | Method | BLEU|
> |:----------|:-------:|
> | Transformer| 34.40|
> |+Back-Translation (BT) | 36.38	|
> |+Ours  | 36.47 |
> |+BT, +Ours | 37.17|
>
> As shown in the table, when jointly using BT and our method, the BLEU score (37.17) is better than using any one of them solely (36.38 or 36.47). The result exemplifies that our method is complementary to other data augmentation methods. We are also working on the combination with other methods and plan to share them when ready.
>
> > 2. Question about the computation cost
>
> One advantage of our method is that we don’t have extra costs during inference. We evaluate the average training time on 4 Titan XP GPUs. The extra cost is only 32 seconds on every epoch, which is 33% of one training epoch of the baseline. The extra time is smaller than other works.  Sparsifying the softmax is also a valuable suggestion which we plan to try in our future work.
>
> > 3. Question about the For the typos and grammar issues
>
> We feel grateful for finding those issues and have changed them in the revised version.
>
> > 4. Real-world usefulness compared with back-translation
>
> Back translation is a widely used method in real-world machine translation tasks. As we have shown in the previous table, our method is complementary to back-translation. In addition, back-translation may not be efficient on the sequence generation tasks which do not have duality because of the information loss. However, our method is still useful in these tasks.
>
> > 5. Question about the relation of temperature and vocabulary size
>
> We empirically observed the relationship between the optimal temperature and vocabulary size. While formally proving it is mathematically beyond the scope of this paper, we could share some preliminary intuitions here. A higher temperature produces a more "soft" distribution and makes the model pay attention to more tokens. Therefore, when the vocabulary is small, the model takes advantage of the distilled soft distribution and obtains more information. However, when the vocabulary size is large (e.g., CNN/DM), it may be overwhelming for the model to focus on too many tokens. We believe this is an important topic and will study more about this problem in the future.
>
> ----
>
> We hope that these answers can address your questions. If you have future suggestions or questions about our paper, we will feel very happy to share more responses.

---

> > ### Comment · Reviewer_iLjj · 2021-11-28
> > **Reviewer Comment After Response**
> >
> > Thanks for the detailed response and the additional result. It's good to see that the proposed method is stack-able with back-translation. It will be interesting to see how this stacks with other data augmentation methods, but I understand that those results may not be ready within the review discussion period.
> >
> > I do share the other reviewer's concern that the machine translation results are reported with the moses script instead of sacreBLEU -- this will make subsequent reproduction significantly harder. So I will *strongly* encourage the authors to switch to sacreBLEU despite mismatch with previous work. (This should not be hard as long as you still have model outputs so I'd suggest doing this ASAP just to check this off for the other reviewers as well.) Like the other reviewers, I also have slight confusions about the motivation behind the consistency loss, even after reading the discussion, but since this is not the key novelty of this paper and ablation studies have been conducted, I'm fine with just empirical investigations.
> >
> > I'm updating my overall evaluation to accept.

---

### Decision · Program_Chairs · 2022-01-20

**Decision:**

Accept (Poster)

**Comment:**

This paper presents a method for target side data augmentation for sequence to sequence models.  The authors of the paper use a relatively straightforward method to generate pseudo tokens that are used for enhanced training.  The authors present results on dialog generation, MT and summarization where automatic metrics show improvements.  For really robust results, I would have preferred to see more human evaluations since BLEU and ROUGE are metrics that the NLP community is moving away from.  Overall, the majority of the reviewers are happy with the paper and there is significant back and forth between the reviewers and authors that have improved the paper;  I think the authors went to significant lengths to allay all concerns from the reviewers and the paper should be accepted.